# The Influence of *Viburnum opulus* Fruits Addition on Some Quality Properties of Homogenized Meat Products

Monika Mazur [1,*], Anna Marietta Salejda [1], Kinga Maria Pilarska [1], Grażyna Krasnowska [1], Agnieszka Nawirska-Olszańska [1], Joanna Kolniak-Ostek [1] and Przemysław Bąbelewski [2]

[1] Faculty of Biotechnology and Food Science, Wrocław University of Environmental and Life Sciences, 37 Chełmońskiego Str., 51-630 Wrocław, Poland; anna.salejda@upwr.edu.pl (A.M.S.); kinga.pilarska@upwr.edu.pl (K.M.P.); grazyna.krasnowska@upwr.edu.pl (G.K.); agnieszka.nawirska-olszanska@upwr.edu.pl (A.N.-O.); joanna.kolniak-ostek@upwr.edu.pl (J.K.-O.)

[2] Faculty of Life Sciences and Technology, Wrocław University of Environmental and Life Sciences, 24A Grunwaldzki Sq., 53-363 Wrocław, Poland; przemyslaw.babelewski@upwr.edu.pl

* Correspondence: monika.mazur@upwr.edu.pl

**Abstract:** This review describes the effect of added freeze-dried guelder rose fruit powder (GRFP) on the quality parameter of homogenized meat products (HMP). In this study, the pH, cooking loss, instrumental color (L*, a* and b*), texture profile, TBARS (Thiobarbituric acid reactive substances), antimicrobial assay, polyphenol content and sensory evaluation of HMP were evaluated. Due to the antioxidant activity of guelder rose fruit (the chlorogenic acid dominated among the identified compounds of the fruit), it was observed that, in the samples with the highest amount of the extract, the amount of microorganisms responsible for food spoilage decreased after storage time (14 days, 4 °C). The addition of lyophilized fruits with a low pH value resulted in the highest cooking loss. The results of sensory evaluation show that, along with the increase of GRFP addition, the taste and smell of meat products become less acceptable. The bitter taste of guelder rose fruit can affect the sensory assessment of meat products. This study is of a pilot nature; further research attempts will be made to offset the problems and design certain solutions, e.g., the use of freeze-dried encapsulation and its addition to meat products.

**Keywords:** meat products; guelder rose fruit; chlorogenic acid; lipid peroxidation; color; phenolic acids; *Viburnum opulus*

## 1. Introduction

*Viburnum opulus*, known as European guelder, is also termed, e.g., Guelder rose, European cranberrybush, wild guelder rose and, in Turkey, gilaburu [1] *V. opulus* belongs to the Caprifoliaceae family and has extensive coverage, including Western and Central Europe, through Asia, Caucasus to Asia Minor. Guelder rose grows up to 3 or 4 m on moist and fertile soils deciduous forests. Its stems are straight and arched, overhanging with age. Guelder rose makes fruits after flowering (snow-white flowers developing at the break of May and June). They are collected in infructescence made of several to tens fruits. At the beginning fruits are green and with maturation they turn red at the end of September, at full maturity, they are red and shiny.

Guelder rose is most often found in a wild and secondary wild form in local parks. Despite the esthetic value of *V. opulus*, the fruits in some of Eastern countries are used for preparing jam, marmalades, juices, pies and herbal teas [1,2]. In Canada, they are used as a cranberries. The unpleasant, bitter taste of the fruits is due to the fact that they contain saponin glycosides and vinburnine, which are considered slightly toxic and require freezing before eating [3].

Guelder rose is a popular shrub in Poland; however, despite its nutritious qualities, it has become a forgotten fruit, rarely used in cooking or in natural medicine. In the past,

its fruits and fruit juice have been used in herbal medicine, for example in treatment of coughs and cold, neurosis, diabetes, bleeding, heart disease, high blood pressure and endometriosis in women [1,3,4]. The use of guelder rose fruits on an industrial scale is actually very limited. In Poland, the fruit is disregarded, but in Turkey, there is a widely popular gilaburu juice, which is a traditional non-alcoholic fermented beverage that is produced commercially. It is popular and widely described in many publications due to its functional properties [5]. There are many publications which have demonstrated the possibility of using the fruit's antimicrobial properties in the textile industry and as an antimicrobial agent in food products. Many of the publications revealed the potential of the fruit's successful application on an industrial scale. Nowadays, on the Polish shore, one can buy the bark of *V. opulus*, as well as shrubs for planting in the garden.

*V. opulus* fruit contain biologically active compounds, such as phenolic compounds (including phenolic acids, anthocyanins and chlorogenic acids), organic acids (including ascorbic and L-malic acids), carotenoids, triterpenes, iridoids, essential oils, saponins and dietary fiber [1]. It has been reported that the compounds included in the fruits improve lipid and carbohydrate metabolism and have anti-inflammatory and anti-thrombotic properties, as well as antitumor activity [5–7]. Due to its attractive color and the fact that this wild shrub has a high content of bioactive components, such as phenolic compounds, it was decided to use *V. opulus* (its freeze-dried form) as an ingredient in the recipe for homogenized meat products. The chosen products are of a model nature, while the research comprises pilot studies. To date, guelder rose fruits have been examined for antioxidant properties, chemical composition and microbial properties in cell lines. The addition of *V. opulus* to meat products is a new method that has not yet been described in literature. The planning of the experiment required numerous issues to be resolved before the research could start, such as the influence of fruit addition on the taste and smell of the finished product, the yield of the final product resulting from the high acidity of the added raw material and the loss of healthy compounds during food processing.

This study will bring new solutions to this type of research. It is also a part of contemporary research that concerns the application of natural additives—bioactive and healthy compounds—to meat products. Further research of this type will pursue the use of natural derivative additives, which are also a good alternative to synthetic additives. Such a solution may be enthusiastically received by consumers.

Due to consumers' interest in high-quality food, the producers focused on functional meat products with natural ingredients. Enriching meat products with bioactive compounds, e.g., herbs and fruits, results in obtaining functional properties and can also be a source of bioactive compounds that have beneficial effects on human health.

The main purpose of using natural additives in meat is to shape its quality and eliminate such crucial problems as lipid oxidation and development of pathogenic microflora in meat products, which result in economic loss, lowers product quality as well as negatively affects consumer experience [8].

## 2. Materials and Methods

### 2.1. Plant Material

Geulder rose fruit (Figure 1) were grown in the collection of shrubs and trees of the Department of Horticulture of Wroclaw University of Environmental and Life Sciences (Poland). Shrubs were cultivated in spacing 2 on 3 m on soils called degraded template, according to soil classification. Soil surface was covered with 5 cm layer of pine bark. Ripe and frozen fruits were harvested in December 2018 and stored for one week at −18 °C. After removing the seeds, the fruit was grinded in a mortar. The pulp was poured onto plates and freeze-dried for 2 days (LABCONCO Corporation, Kansas City, MI 64132, USA). The lyophilized fruits were vacuum sealed and stored at −18 °C until use.

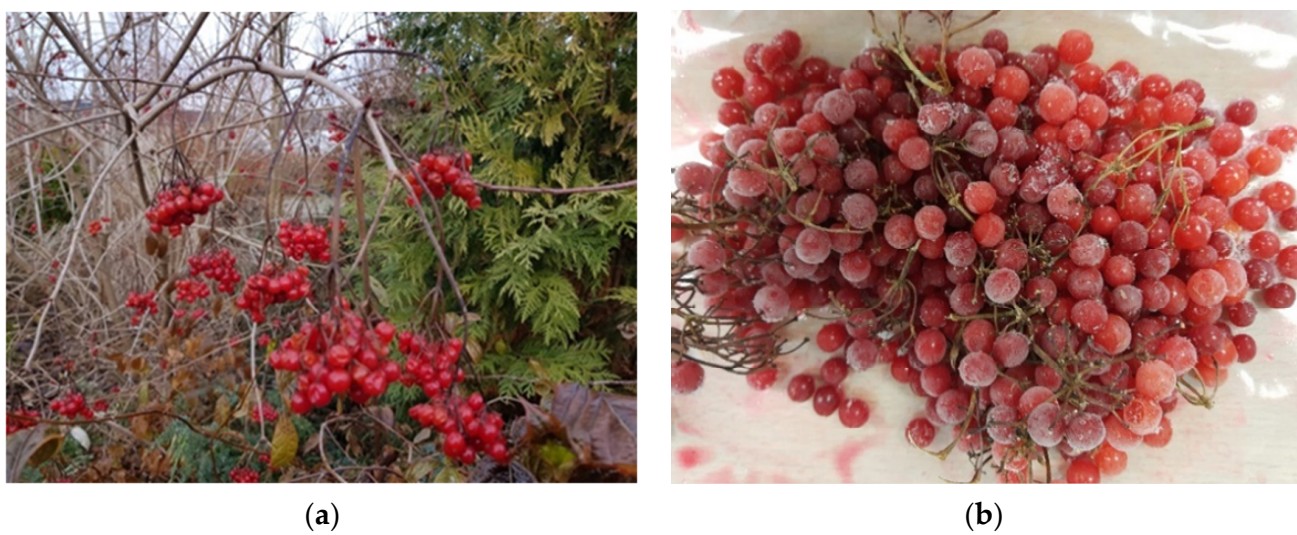

(**a**)                                        (**b**)

**Figure 1.** Guelder rose shrub. Author P. Babelewski (**a**) Guelder rose fruit after freezing. Author M. Mazur (**b**).

### 2.2. Preparation of Homogenized Meat Products

The experimental model consisted of four production treatments: A (control sample) and B, C, D produced with 0.05, 0.1, 0.15 g/100 g of GRFP (freeze–dried guelder rose fruit powder) addition, respectively (Table 1).

**Table 1.** Homogenized meat products recipe (g/100 g).

| Ingredients g/100 g | A | B | C | D |
|---|---|---|---|---|
| Pork | 50.00 | 50.00 | 50.00 | 50.00 |
| Backfat | 30.00 | 30.00 | 30.00 | 30.00 |
| Ice | 20.00 | 19.95 | 19.90 | 19.85 |
| Curing salt | 1.2 | 1.2 | 1.2 | 1.2 |
| Concentrate of spices | 0.6 | 0.6 | 0.6 | 0.6 |
| Sugar | 1.0 | 1.0 | 1.0 | 1.0 |
| GRFP | 0.00 | 0.05 | 0.1 | 0.15 |

Samples: A, B, C, D, GRFP—freeze–dried guelder rose fruit powder.

Grounded raw materials (post-rigor pork cuts (*m. semimembranosus*) and backfat) together with all ingredients were homogenized at 9000 rpm (Büchi Mixer B-400, BÜCHI Labortechnik GmbH, Deutschland). Approximately 60 g of stuffing was packed into polypropylene tubes (2.5 cm diameter and 12 cm height), sealed and cooked in water bath (Julabo TW12, Julabo Inc., Allentown, PA, USA) until the temperature of geometric center of sample reached 72 °C. After, the final products were cooled down on ice to 21 °C, vacuum packed and stored for 14 days at 4 ± 1 °C. Two independent production batches were carried out. Quality properties of homogenized meat products (HMP) were assessed immediately after the production process and after 14 day of chilled storage.

### 2.3. PH Value

The pH value was measured by using Orion 3-Star pH meter (Benchtop pH Meter, Thermo Fisher Scientific Inc., Waltham, MA, USA) at room temperature. Before each measurement, the samples of HMP were fragmented and tightly packed in flasks. The pit of electrode was inserted directly to such prepared samples.

### 2.4. Weight Losses

Weight losses were measured as cooking losses during thermal treatment of homogenized meat products and were expressed as a percentage of initial sample weight.

### 2.5. Color Measurement

Color of homogenized meat products surface was measured directly after production process and after 14 d of chilled storage by using a reflectance colorimeter Minolta CR-400 (Konica Minolta, Osaka, Japan). Color values presented in this paper were expressed in CIE Lab system (L*—lightness, a*—redness, b*—yellowness). Samples of HMP were cut crosswise into 10–15 mm thick slices before each measurement. For each treatment, six measurements were made.

### 2.6. Texture Profile Analysis

Texture profile analysis (TPA) of homogenized meat products was evaluated in Zwick/Roell Z010 testing machine (Zwick Testing Machines Ltd., Leominster Herefordshire, UK). Slices of HMP (10–15 mm thick) at day 0 were compressed twice to 50% of their original height. The head speed 60 mm/min and relaxation time 30 s were set. Textural parameters such as hardness, cohesiveness, springiness, chewiness and gumminess were measured.

### 2.7. TBARS

Intensity of oxidation processes was measured by the determination of reaction products with thiobarbituric acid (TBARS, Thiobarbituric acid reactive substances). The content of malondialdehyde (expressed as mg MDA/kg of sample) was measured according to the method proposed by Pikul et al. [9] and Mei et al. [10], with our own modification. Consequently, 0.5 g ± 0.01 g of HMP and 10 mL of 10% TCA was placed in centrifuge tube, homogenized and centrifuged at room temperature for 10 min (4000 rpm, Sigma 3K30, Sigma Laborzentifugen GmbH, Osterode am Harz, Germany). A total of 2 mL of supernatant was treated with 2 mL 0.02 M aqueous solution of TBA and heated at 100 °C for 40 min. The samples were cooled down on ice and measuring absorbance on a UV-1800 spectrophotometer (BRAIC, China) at 532 nm versus blind test (supernatant was replaced by TCA solution). The measurements were carry out four times.

The malondialdehyde content (Y) was calculated using the formula obtained from the standard curve:

$$X = Y \times 0.0118 - 0.009$$

X—absorbance value at wavelength $\lambda = 532$ nm.

### 2.8. Antimicrobial Assay

The content of microorganisms in the samples were assessed by using the plate method. A total of 6 media selective for specific groups of microorganisms were used: PCA, Plate Count Agar (Merck Millipore, USA), YGC, Yeast Extract Glucose Chloramphenicol (BTL Sp. z o.o Łódź, Poland), MRS, de Man, Rogosa i Sharpe(BTL Sp. Zoo, Łódź, Poland), McConkey (BD GmbH, Heidelberg, Germany), BACARA ® (Biomerieux, Warsaw, Poland), ChromiD (Biomerieux, Poland). Samples were taken sterile (5 g each), placed in sterile bags and homogenized with 45 mL of sterile physiological fluid (0.85% NaCl) for 2 min using a Stomacher 400 LAB Blender (Seward Medical, London, UK). Serial dilutions of the samples were prepared and plated on the media to identify the presence of microorganisms and determine their viability. The determinations were made in control samples (without GRF) and tests with the addition of GRFP (B, C, D) after 7 and 14 days of refrigerated storage (4 °C). A total of 120 h of incubation time and temperature 30 ± 1 °C were set. Readings were made manually (colony counter N.USUI and Co. Ltd., BIO KOBE, Japan) after 3, 4 and 5 days of incubation and expressed in CFU per 1 g of test products Antimicrobial assay was performed in duplicate for two batches. Results were averaged for two biological replicates (part 1 and part 2).

### 2.9. UPLC-MS Method for Identyfication of Polyphenols

Extraction of polyphenols was determined using the method described by Püssa et al. [11] with own modifications. The samples of HMP (5.00 ± 0.01 g) extracted with 5 mL of methanol (80%, Sigma-Aldrich, Steinheim, Germany) then acidified with HCl. Next, the samples were shook for 30 min at room temperature and centrifuged (5000 rpm, 10 min.). After the addition of hexane (10 mL) to supernatant, the hydrophilic layer was obtained and kept in Eppendorf vials at 18 °C until analysis. Before the identification the sample diluted with 0.1% formic acid (1:1 *v/v*, Sigma–Aldrich Steinheim, Germany). For the identification of polyphenols, the protocol proposed by Kolniak-Ostek [12] was used. The identification of polyphenols in HMP extracts was carried out using an ACQUITY Ultra Performance LC system equipped with a photodiode array detector with a binary solvent manager (Waters Corporation, Milford, MA, USA) with a mass detector G2 Q-Tof micromass spectrometer (Waters, Manchester, UK) equipped with an electrospray ionization (ESI) source operating in negative mode. For the separation of single polyphenols, an UPLC BEH C18 column (1.7 mm, 2.1 × 100 mm, Waters) was used. The mobile phase consisted of 0.1% formic acid, *v/v* (solvent A) and 100% acetonitrile (solvent B).

The single components of polyphenols were characterized based on the retention time and the accurate molecular masses. The data obtained from UPLC–MS were analyzed in the MassLynx 4.0 ChromaLynx (Application Manager software, Waters).

Phenolic acids were monitored at 320 nm and flavonol glycosides at 360 nm. The PDA spectra were measured over the wavelength range of 200–600 nm in steps of 2 nm. The retention times and spectra were compared to those of the authentic standards (chlorogenic acid, quinic acid and apigenin di-glucoside were purchased from Extrasynthese, Lyon, France).

### 2.10. Sensory Evaluation

The Polish standard PN-ISO 4121:1998 [13] was used for sensory evaluation. The HMPs were prepared for serving using the following steps: cooled down (up to 21 °C), cut (1 cm slices) and evaluated using a hedonic rating scale of acceptance (scale: from 1—extremely dislike to 9— extremely like) [14]. The parameters of overall appearance, color, smell, taste and hardness were investigated.

### 2.11. Statistical Analysis

For the statistical analysis, Statistica software ver. 8.0. (StatSoft Inc., Poland) was used. The results (mean ± standard error) were examined using Duncan's multiple range test ($p \leq 0.05$) to identify the statistically significant effects.

## 3. Results and Discussion

The pH value has an impact on many quality parameters, such as water holding capacity, tenderness, color, juiciness, flavor, aroma, protein proteolysis concentration and shelf life. The pH of a meat product was measured after production (day 0). The pH values of the samples varied from 5.95 to 5.78 (Table 2). Differences in pH values across samples with GRFP addition were observed. Samples with GRFP were added, characterized by lower pH value than control (A). This indicates that the addition of GRFP significantly decreases pH values. This can be attributed to the accumulation of acidic products from the added fruits. A similar effect was observed in the study Radha et al. [15], which indicated that the addition of dried cloves, cinnamon, oregano and mustard extracts to raw chicken meat decreased the pH values of samples during storage. According to the authors, this effect could be connected with growth of bacteria (bacteria's metabolized amino acid, which are released during the breakdown of proteins). Cam et al. [16] showed that a low guelder rose fruit's pH is 2.95. Our study indicated that GRFP's total acid content amounts to 1.78 g/100 g. Thus, the addition of GRFP affects the pH of meat products in this study. In their study, Yosop et al. [17] observed that, due to the lower value of pH of marinated

poultry meat, the L* value was higher and meat was lighter. In our study, the HMP with a lower pH value (with GRFP addition) was not lighter.

**Table 2.** The values of pH and cooking loss of HMP (homogenized meat products).

|   | pH | Cooking Loss |
|---|---|---|
| A | 5.95 [a] ± 0.01 | 8.71 [a] ± 0.68 |
| B | 5.87 [b] ± 0.02 | 26.14 [b] ± 0.5 |
| C | 5.80 [c] ± 0.01 | 26.09 [b] ± 2.29 |
| D | 5.78 [c] ± 0.01 | 26.24 [b] ± 2.04 |

[a,b,c]—different letters indicate significant differences ($p \leq 0.05$) between values. A—control sample, B, C, D samples produced with 0.05, 0.1, 0.15 g/100 g of GRFP, respectively.

The cooking loss of control samples was determined at 8.71% and it was significantly ($p \leq 0.05$) lower than in the experimental samples containing lyophilized GRFP (26.14–26.24%). The samples (B, C and D) were no different in their values of cooking loss. An increase of cooking loss, along with increasing addition of the plant component, was also observed in the study conducted by Barbieri et al. [18]. The addition of Polygonum cuspidatum and rosemary extract (dose 148 mg/kg, expressed as gallic acid equivalent) caused cooking losses determined to be 26.85%.

The addition of lyophilized fruits with a low pH value results in the highest cooking loss. This occurrence a consequence of the protein denaturation, which caused the decreased ability of meat emulsion to bind water and fat. Cooking loss is an important issue that concerns the quality of meat, as well as economic aspects [19]. In further research, attempts will be made to offset the problem of high cooking loss and design certain solutions, e.g., the use of freeze-dried encapsulation and addition to meat products.

The applied additives ($p \leq 0.05$) resulted in a decreased value of profile texture parameters (Table 3). A significant dependence was found between the level of the addition of GRFP and gumminess and chewiness. Meat products with GRFP were characterized by lower gumminess and chewiness than those in the control sample (A). Some studies found meat samples immersed in acid marinades characterized by lower hardness than control samples, which may be related to a decrease of water–binding capacity of meat proteins [17]. However, in our study, the hardness values of HMP samples produced without GRFP and samples produced with the highest addition GRFP did not differ statistically. Further research is needed to draw conclusions and possibly address discrepancies in results, as well as to learn about the relation between guelder rose fruit constituents and meat batter ingredients.

**Table 3.** Values of the TPA (texture profile analysis) parameters for HMP.

|   | Cohesiveness [-] | Springiness [mm] | Hardness [N] | Gumminess [Nm] | Chewiness [N] |
|---|---|---|---|---|---|
| A | 0.55 [a] ± 0.05 | 0.82 [a] ± 0.25 | 30.06 [a] ± 2.4 | 19.31 [a] ± 1.83 | 25.22 [a] ± 2.66 |
| B | 0.42 [b] ± 0.15 | 0.76 [a] ± 0.21 | 28.0 [b] ± 6.26 | 11.06 [b] ± 5.83 | 19.68 [b] ± 2.27 |
| C | 0.34 [c] ± 0.11 | 0.61 [c] ± 0.17 | 31.0 [a] ± 0.84 | 5.59 [c] ± 1.35 | 8.06 [c] ± 2.74 |
| D | 0.4 [b] ± 0.05 | 0.68 [c] ± 0.07 | 29.4 [a] ± 3.21 | 10.13 [d] ± 2.74 | 16.53 [d] ± 2.07 |

A—control sample, B, C, D samples produced with 0.05, 0.1, 0.15 g/100 g of GRFP, respectively, n = 10; Day 0, [a,b,c]—different letters indicate significant differences at $p \leq 0.05$ between values.

The results of color parameters (average values) are shown in Figure 2. In control sample A, the highest mean L* values were observed and they increased during the storage period from 77.12 ± 0.83 (day 0) to 79.04 ± 2.71 (day 14). In a similar effect obtained in the study by Fernández-López et al. [20], the control samples showed higher L* values than samples treated with rosemary and hyssop extracts.

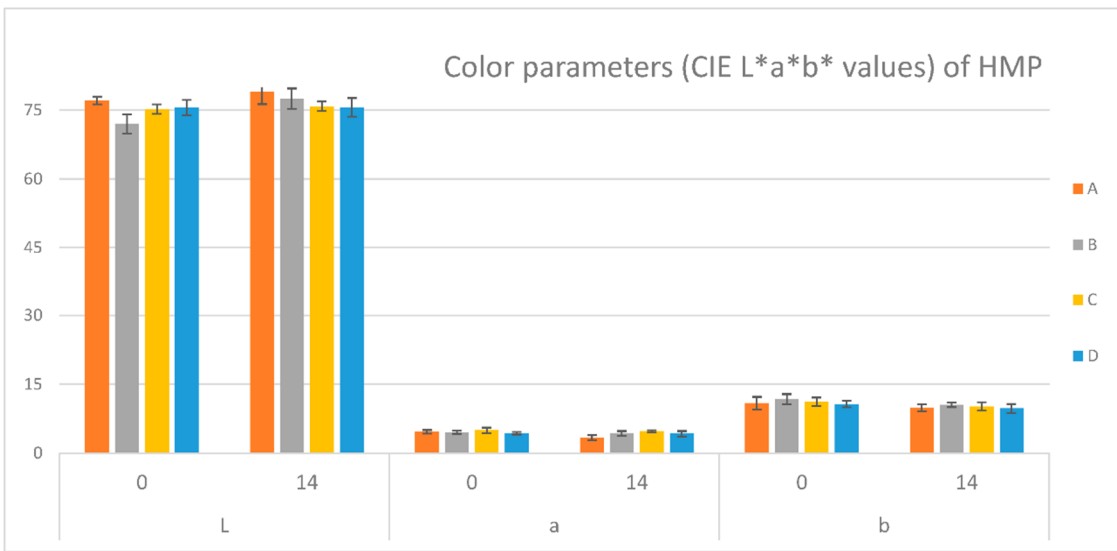

**Figure 2.** Color parameters (CIE L*a*b* values) of HMP measured at day 0 and day 14, *n* = 6. A—control sample; B, C, D samples produced with 0.05, 0.1, 0.15 g/100 g of GRFP, respectively.

Different results were obtained for the variants with the addition of GRFP (C, D). After 14 days, the value of the L* parameter did not differ statistically from day 0 and its value was lower than in sample A without GRFP added. Regarding the variation of L*, it was observed during the entire experimental period that the values of all samples (A, B, C and D) decreased. The addition of GRFP resulted in a higher stability of the L* parameter between day 0 and day 14 than in the control sample. The findings in our study, showing a decreasing trend in L* values, were different than those showed Rooyen et al. [21], which exposed that tendency to increase the L* values. A similar effect to the one in this study (decreased L* value) was found by Oliveira et al. and Fernández-López et al. [20,22]. The decrease of luminosity may be as a result of increasing meta-myoglobin after storage, which was also observed in a higher intense brown color of samples. Moreover, as a consequence of oxidizes the myoglobin to oxy-myoglobin the L* value decreased. In this study, a similar effect to the one in the research conducted by Oliveira et al. was observed, i.e., that higher levels of GRFP added to fresh and stored meat products induced higher color stability. This feature makes the product attractive even if it is stored. One parameter which indicated the red color is a* parameter, which is the most important and sensitive in the context of meat quality. In the results obtained for a* values for the experimental samples, there were no statistically significant differences at time 0. Nevertheless, a* values of sample A, i.e., $3.35 \pm 0.57$, were lower than those observed in samples B, C and D–$4.26 \pm 0.54$; $4.78 \pm 0.22$ and $4.22 \pm 0.58$, respectively, after a 14-day storage period. The redness of the control sample decreased very rapidly during storage period, from $4.67 \pm 0.39$ to $3.35 \pm 0.57$. The similar effect reported Fernández-López et al. [20], in which a* parameter of the control sample lower from 6.88 to 4.15 after 8 days of storage at 4 °C.

In the samples B, C and D, there were no significant differences ($p \leq 0.05$) in the a* values during the experimental time (day 0 and day 14). The results show that higher a* values were not induced by higher levels of GRFP addition (after 14 days of storage, samples B, C and D were characterized by a more intense color and color stability than control sample A). The study conducted by Draszanowska et al. [23] revealed that the addition of chili peppers to canned meat resulted in a higher value of redness (a*) than in the control samples. Such a correlation could not be observed in our study. The important difference is the lower value of the a* parameter observed for the control sample during storage time. The decrease in redness in cooked meat after the refrigeration period is a typical observation and it is due to the accumulation of lipid oxidation products. Similar results were also found in the study Fernández-López et al. [20]. Guelder rose fruit is rich

in anthocyanins, a strong pigment that adds an intense red color to meat and makes it highly desired by consumers. This effect was noticed in the conducted sensory evaluation. The samples with a higher level of addition of the GRFP are characterized by greater color acceptability [24].

Yellowness index (b* values) indicated crucial changes in quality, e.g., bacterial deterioration or oxidation, which disqualifies the meat for usage. The higher the b* values, the greater the "oxidized/reductive form" ratio. Desired meat products are characterized by a b* value near zero and, there, the yellow and blue colors were not dominated [18,19]. In our study, the b* values of samples A and D did not differ statistically. This trend was also maintained after the storage period and the b* values of all samples on day 14 were lower than the values recorded on day 0.

The amount of MDA gives precious information about lipid oxidation, as well as indicating the quality of meat products. In the TBARS test, MDA is a substance that reacts with thiobarbituric acid. Figure 3 shows average values of mg MDA/kg in meat products measured after production (day 0) and after the storage period (day 14). In the first experimental period (after production), samples A differed significantly from samples B, C and D (formulation prepared with GRFP added). In the formulation with no GRFP, a higher amount of MDA than in the group with GRFP was mentioned. Moreover, sample D contained the highest amount of GRFP (0.15 g/100 g), was characterized by the lowest amount of MDA and displayed the highest resistance to lipid peroxidation. However, a different relationship was observed in studies conducted by Ryu et al. [25]. The lower addition (0.5%) of grape skin and seed pomace (GRS) decreased the levels of TBARS more efficiently than the higher addition (1%) of GRS. During the storage period (4 °C, 14 days), higher values of MDA in comparison to the values measured on day 0 in all samples were observed, an effect of lipid peroxidation. Based on the average TBARS values of experimental samples, it is demonstrated that the samples C and D (0.05 and 0.1 g/100 g of GRFP, respectively) had a similar antioxidative effect. Sample A had a significantly higher amount of MDA than other samples during the entire storage period, whereas sample D was more efficient in terms of the antioxidant level than in other groups.

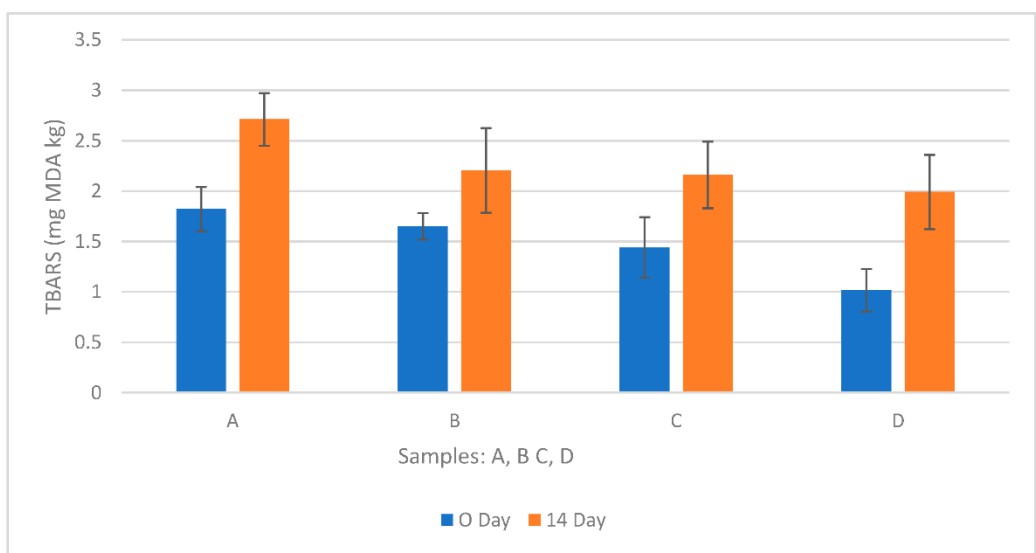

**Figure 3.** TBARS levels (average mg MDA/kg value) of fresh and stored meat products prepared with different levels of GRFP, A—control sample; B, C, D samples produced with 0.05, 0.1, 0.15 g/100 g of GRFP, respectively.

In the conducted study, the levels of TBARS in all samples increased during the storage period. The same relation has also been shown in many studies [22,24,26].

In our study, the levels of MDA are between 1.017–2.711 and it is possible to detect rancidness in meat products. Lipid peroxidation becomes detectable by senses when the

mg MDA/kg value is equal to or higher than 2 mg/kg. The study needs to be repeated, since the change in the MDA value (to a higher one) is not entirely transparent.

No pathogenic microorganisms of *Bacillus cereus* and *Salmonella* species were detected in the tested samples.

It was observed that the HMP produced with GRFP addition were characterized by the lower number of microorganisms: the microbial inhibition grew with increasing the content of GRFP in the recipe of HMP (Figures 4–11, results were averaged for two biological replicates: part 1 and part 2). Moreover, it was observed that, in the samples with the highest amount of the extract, the amount of microorganisms responsible for food spoilage decreased after 14 days of cold storage. This level of addition has a positive effect on increasing shelf life. According to the study by Coman et al. [27] some red fruit extracts indicate antimicrobial properties, especially against the pathogens *B. cereus*, *S. aureus* and *E. coli*. The activity of plant extract indicates the interaction between polyphenols and other bioactive compounds, which depends on many factors, e.g., the method of extraction or part of the plant. Some studies indicate that the mixture of phenolic compounds in plant extract has stronger antimicrobial activity than single extract [28]. The results obtained in the present research indicate that the addition of GRFP can be successfully used to inhibit the growth of microorganisms. Secondary metabolites of *Viburnum opulus*, such as phenolic compounds, as well as metal ions, especially copper and zinc, play a role as antimicrobial agents. The results of Sun et al. [29] indicated that chlorogenic acid are the dominant phenolic compounds in *V. opulus*, which indicated antibacterial activity against *S. Enteritidis*. There are many studies confirming the antimicrobial effect of guelder rose fruits [30,31]. Česonienė et al. [31] noted that juice of *V. opulus* fruits can strongly prevent against human pathogenic bacteria, e.g., Gram-negative: *Salmonella typhimurium* and *S. agona* and Gram-positive *Staphylococcus aureus*, *Listeria monocytogenes* and *Enterococcus faecalis*.

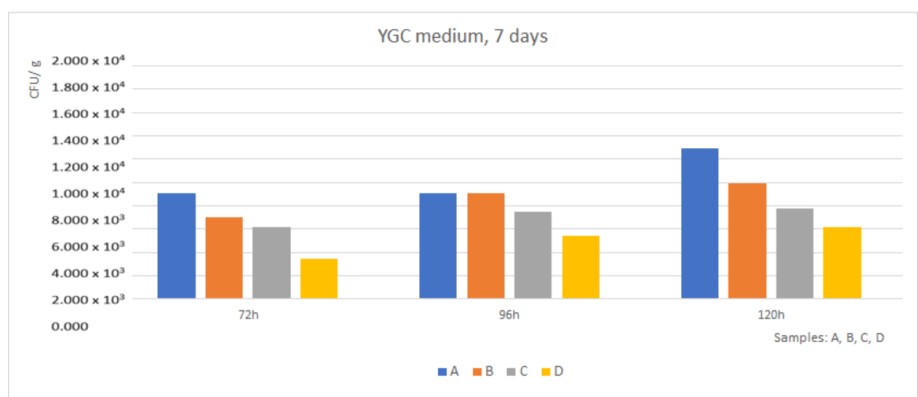

**Figure 4.** YGC -medium (Yeast Extract Glucose Chloramphenicol Medium), day 7.

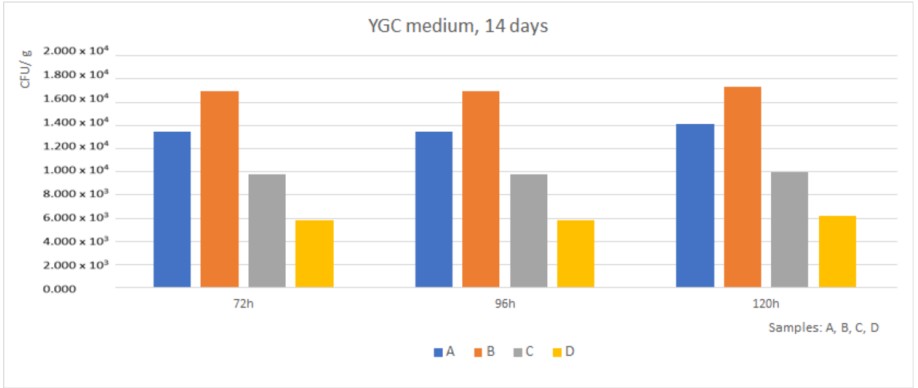

**Figure 5.** YGC – medium (Yeast Extract Glucose Chloramphenicol Medium), Day 14.

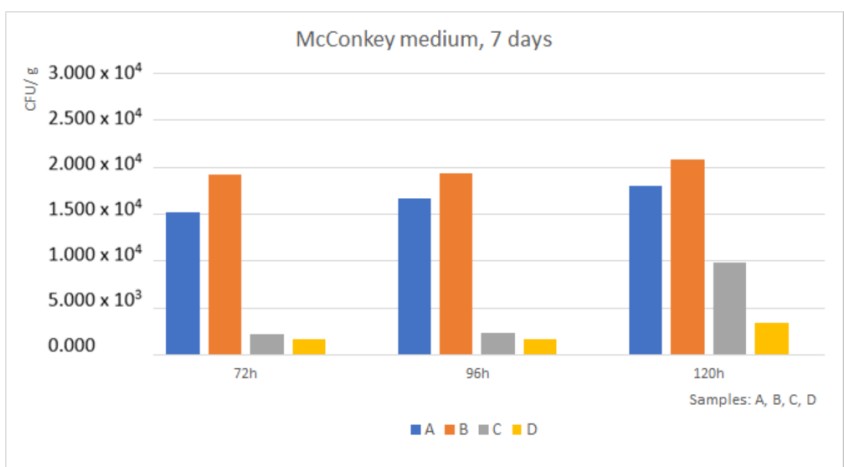

**Figure 6.** McConkey medium, day 7.

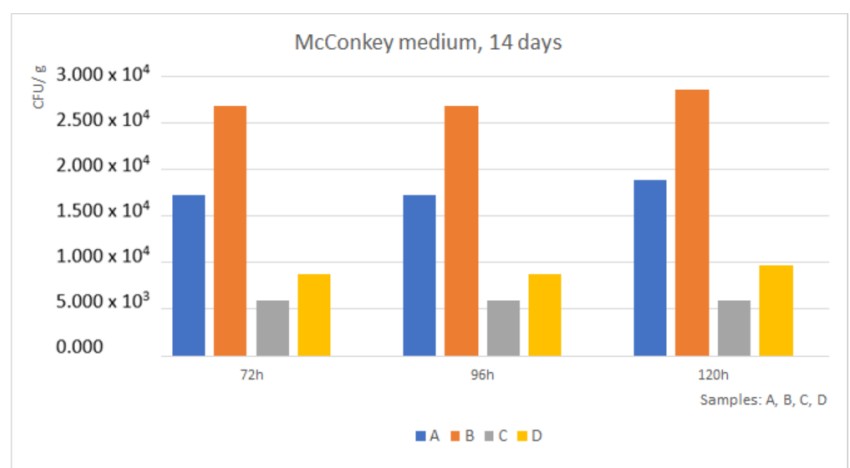

**Figure 7.** McConkey medium, day 14.

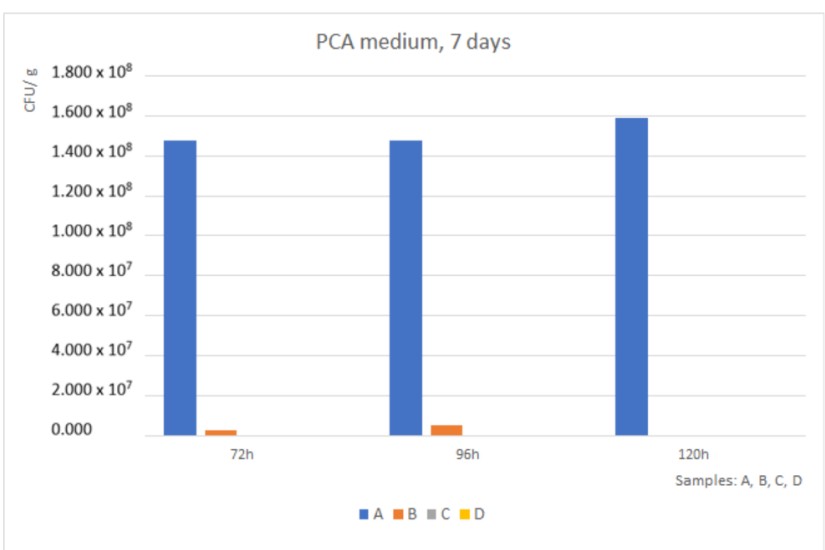

**Figure 8.** PCA medium (Plate Count Agar), day 7.

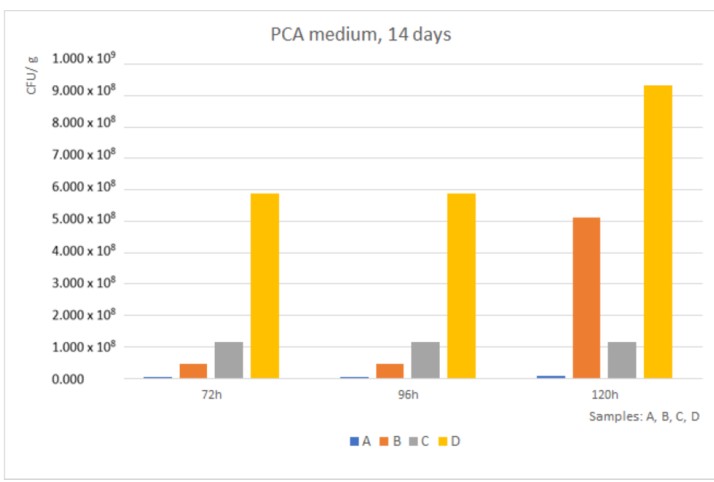

**Figure 9.** PCA medium (Plate Count Agar), day 14.

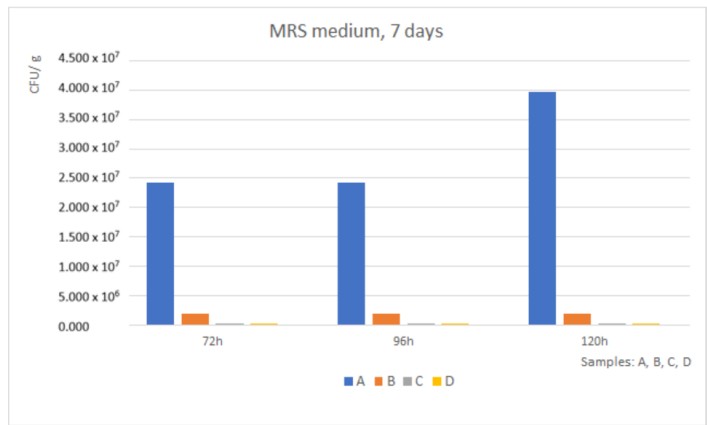

**Figure 10.** MRS medium, day 7.

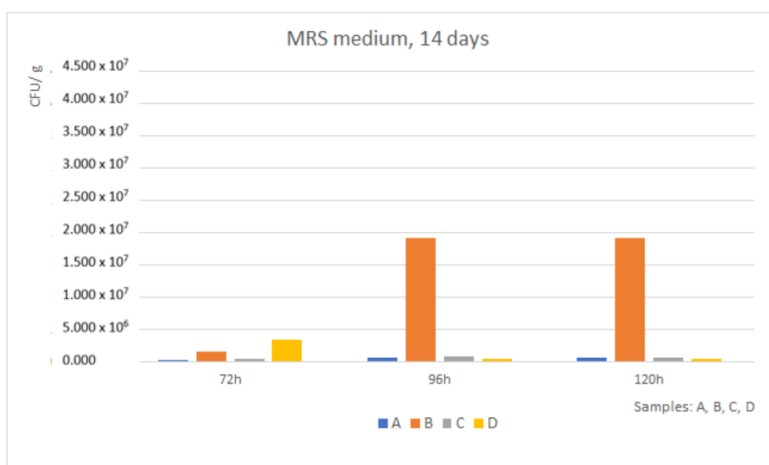

**Figure 11.** MRS medium, day 14.

Table 4 presented the results of the UPLC-MS analysis. As a result of qualitative identification in HMP, components were obtained such as chlorogenic acid, quinic acid, 5,7-dihydroxyflavone (chrysin) and 4′,5,7-trihydroxyflavone (apigenin) [32]. A few authors have studied the phenolic profile of *V. opulus* fruit. Velioglu et al. [33] indicated chlorogenic acid as the dominant component in fresh *V. opulus* fruits (204 mg/100 g),

while Polka et al. [1] demonstrated that chlorogenic acid is not the dominant component in fresh, but in dried fruits (DW, dried weight) is 752.59 mg/100 g. Chlorogenic acid (peak 4, Figure 12) were the major identified compounds contributed to antioxidant activity and have positive health effects [34]. The non-nutrient plant compounds, such as phenolic acids, play a significant role in meat products, such as delayed lipid and protein oxidation, inhibition of microbial growth and improved color stability [35].

**Table 4.** The phenolic compounds identified in homogenized meat product with GRFP addition.

| Peak Number | λ nm | fr (Analog-PDA) | tMS1 | tMS2 | [M-H]− (*m/z*) | MS/MS Parent Ion | [M-H]+ | Compound | A | B | C | D | Refs. |
|---|---|---|---|---|---|---|---|---|---|---|---|---|---|
| 1 | | 1.15 | | | | | NI | | | | | | |
| 2 | | 1.32 | 1.361 | 1.378 | | 377.262 | | NI | | | | | |
| 3 | | 5.18 | 5.271 | 5.288 | | 375.3049 | | NI | | | | | |
| 4 | | 5.91 | 5.945 | 5.962 | 353.0879 | 191.0548 | | Chlorogenic acid | - | X | X | X | [32] |
| 5 | | 6.16 | | | | | NI | | | | | | |
| 6 | | 6.85 | 6.909 | 6.925 | 191.2798 | | | Quinic acid | - | X | X | X | [32] |
| 7 | 320 | 7.46 | 7.529 | 7.546 | 253.2655 | | | Chrysin 5,7-dihydroxyflavone | - | X | X | X | [32] |
| 8 | | 8.93 | 8.958 | 8.975 | 269.2577 | | | Apigenin 4′,5,7-trihydroxyflavone | - | X | X | X | [32] |
| 9 | | 13.97 | 13,487 | 13,504 | | 265,3621 | | NI | | | | | |
| 10 | | 15.22 | | | | | NI | | | | | | |

NI: could not be identified.

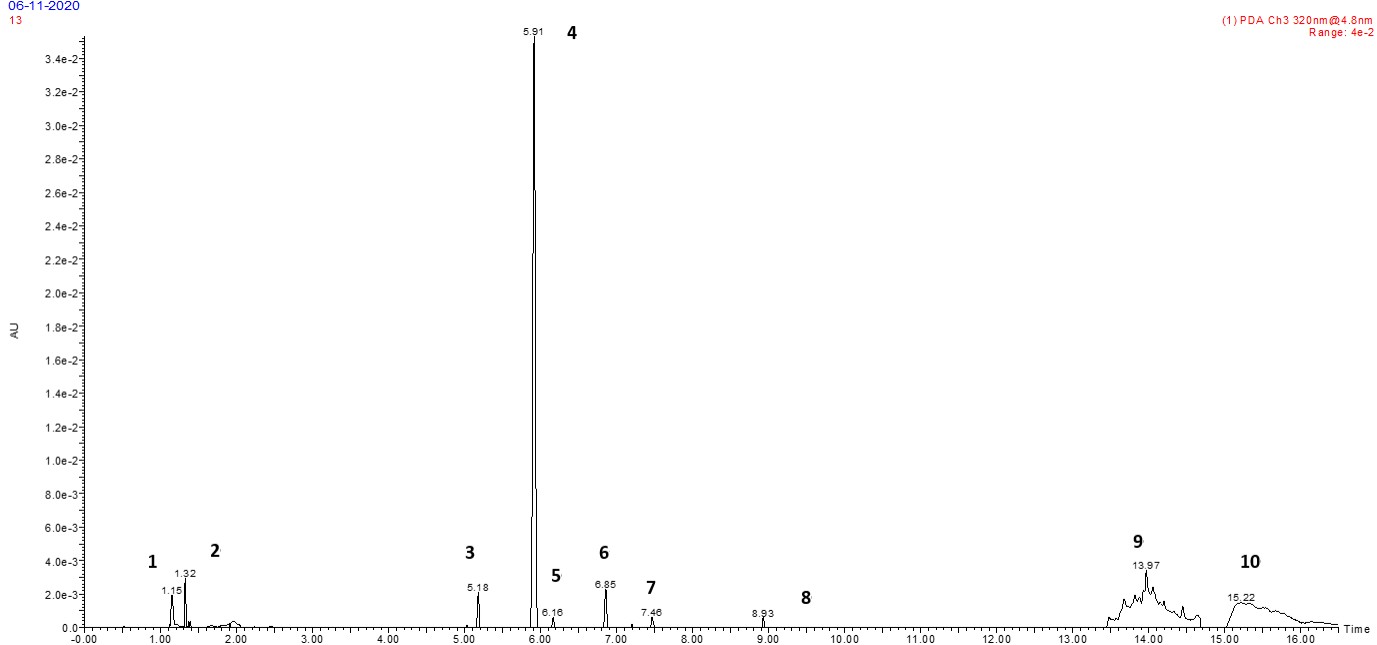

**Figure 12.** Representative UPLC-MS chromatogram of HMP with the highest level of GRFP addition (D sample, 0.15 g/100 g). Chromatogram was obtained at 320 nm. The compound numbering is listed in Table 3.

The results of the consumer assessment have shown that addition of GRFP in the recipe of HMP contribute to more sensory acceptance in overall appearance and color than control samples (Figure 13). Among samples manufactured with GRFP preparation, the highest consumer acceptability of taste was observed for the control sample. Increasing addition of GRFP negatively influenced assessment of smell and taste. Lower acceptability of taste might be related to the bitter taste of guelder rose fruits. However, with the increase of GRFP addition, the taste and smell of meat products, lost its acceptability. The bitter taste of guelder rose fruits could affect the results of the sensory assessment of meat products.

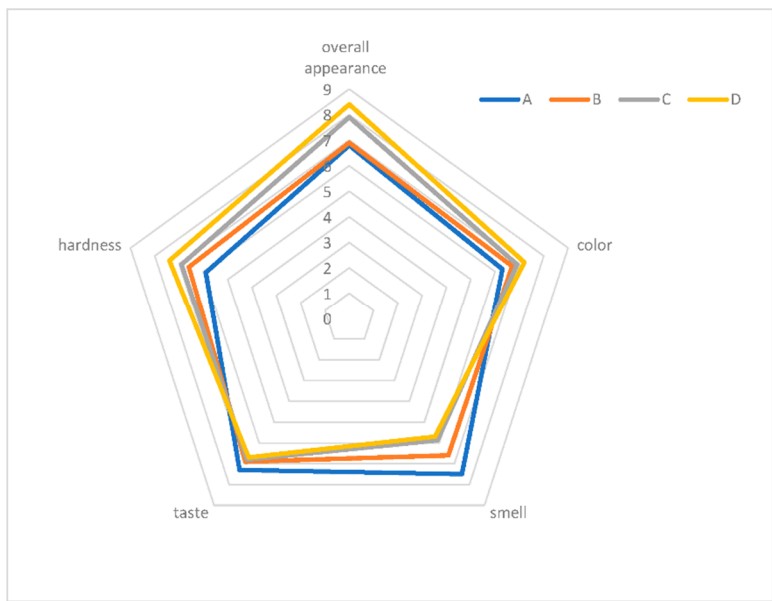

**Figure 13.** Sensory evaluation of HMP with GRFP addition. Samples: A (control sample); B (0.05 g/100 g); C (0.1 g/100 g); D (0.15 g/100 g).

## 4. Conclusions

The trend of adding natural antioxidants to meat products is well promoted among scientists. The concept of our study was to use an interesting fruit with antimicrobial and antioxidant properties. The uniqueness of our research lies in the fact that guelder rose is a forgotten fruit in its wild form and is a subject of numerous scientific publications. In our study, despite the adverse effect, i.e., a significant cooking loss, we demonstrated that the addition of GRFP is an interesting option for improving the microbiological quality of meat products. Moreover, during the refrigeration period (14 days), the samples with the addition of GRFP were characterized by a more intense and stable color than the control samples. The parameters of texture require further research to analyze the changes, while the TBARS test study needs to be repeated, since the change in the MDA value (to a higher one) is not fully transparent. In the study, it was identified that chlorogenic acid, i.e., a strong antioxidant agent, adds antioxidant properties to the products and contributes to the extension of the storage period. Our study is an innovative one and provides an opportunity to use the *Viburnum opulus* fruit as an antimicrobial agent in meat production. Therefore, it is essential that further research is conducted to understand some trends of GRFP activities in HMP, as well as to reduce the adverse effects.

**Author Contributions:** Conceptualization, M.M.; methodology, A.M.S.; G.K.; J.K.-O.; A.N.-O.; P.B.; formal analysis, M.M.; K.M.P.; writing—original draft preparation, M.M.; writing—review and editing, A.M.S.; supervision, A.M.S.; G.K. All authors have read and agreed to the published version of the manuscript.

**Funding:** This research was funded by subsidy for maintenance and development of research potential (Young scientists), grant number B030/0004/1 (Wroclaw University of Environmental and Life Science. Faculty of Biotechnology and Food Science). The manuscript was co-funded by the support project from the subsidy increased for the period 2020–2025 in the amount of 2% of the subsidy referred to Art. 387 (3) of the Law of 20 July 2018 on Higher Education and Science, obtained in 2019.

**Institutional Review Board Statement:** Not applicable.

**Informed Consent Statement:** Not applicable.

**Data Availability Statement:** Data sharing not applicable. No new data were created or analyzed in this study.

**Acknowledgments:** The publication was the result of the activity of the Plants4food research group.

**Conflicts of Interest:** The authors declare no conflict of interest.

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
