# Peer review of "The Influence of Viburnum opulus Fruits Addition on Some Quality Properties of Homogenized Meat Products"

_applsci, doi:10.3390/app11073141_

Round 1

Reviewer 1 Report

Please add more information about the cultivation of viburnum, the availability on an industrial scale of berries.

Line 235. Why you check acids only? It would be better to check pH also.

Tables: use 3 significant figures 

Line 409. bitter cannot be smelled. But it can only be tasted. Please correct.

Figure 9. It is necessary to add statistical analysis.

Please add a more detailed conclusion section.

Reviewer 2 Report

The authors need to follow the following instructions to improve this manuscript.

  • Page 1, Line 20 (Abstract section): spoilage decreased after storage time (14 days): What is the storage temperature?
  • Page 1-2, Line 30-34 (Introduction): Arrange the paragraph. How can make one sentence one paragraph?
  • Page 3, Line 93 (Materials and Methods): Figure 1. GFigure 1.: Please check this
  • Page 3, Line 99: ( semimembranosus): Why italic?
  • Page 3, Line 155: Slices of HMP (10 - 15 mm thick) at Day 0 were compressed twice to: Why Day capital?
  • Page 4, Line 175: X =Y× 0,0118 – 0,009: Check this equation.
  • Page 6, Table 2: What is the meaning/combination of A, B, C, D? Mention as a note.
  • Figure 2: Write the data clearly. Data are not clear. The authors did not mention what they used. Are they used Standard error/deviation? The authors should use a bar in all figures.
  • Table 3: The authors should use 2-digit data in all tables. Sometimes used point and sometimes comma.
  • Result and Discussions: Please compare results with other findings.
  • Page 13, Line 417-420: Conclusions part should improve with the best findings.
  • English Grammar should check by a Native English Speaker or commercial proofreading company.
  • Please check carefully before resubmission.

I recommend to improve the manuscript and resubmit.

Reviewer 3 Report

1.The motivation and purpose of the experiment are interesting, but  revisions are too much.  Thus you should modify and try to submit newly other journal or this journal with great modification such as the following points.

2.According to the literature, we know that polyphenols have antioxidant functions and low pH value can effectively inhibit the growth of microorganisms, so the results of this part are predictable; the results of other analyses are not better than the control group, I think this problem should have been discovered in the preliminary test, and then considered changing the research direction or adjusting the proportion of the recipe. For example, significant cooking loss due to extremely low pH (Line 241).

3.Freeze-drying has high economic costs and requires freeze-drying equipment. Can the marinating method not achieve the goal of the experiment? Please explain. (Line 14)

4.In Table 1, the weight of ice in group C should be 19.90g (20g-0.1g), and the total weight of each group exceeds 100g, please explain? (Line 120)

5.There should be a blank line between Section 2.3 and Section 2.4. (Line 138~139)

6.In UPLC-MS analysis, please explain why quinic acid and apigenin di-glucoside were chosen as standards? (Line 213)

7.Results and Discussion need to cite more literature. (Line 227)

8.The data in Table 2 should be calculated consistently to 2 or 3 decimal places. (Line 253)

9.Figures 2, 4, 5, 6, 7 cannot tell me whether there are significant differences between the groups. (Line 309)

10.There are many errors in the References, please correct them carefully, such as 49, 3, 838-843 --> 49(3), 838-843. (Line 449)

Round 2

Reviewer 2 Report

Please accept cleaned version
